# The Role of Immunotherapy in the Treatment of Advanced Cervical Cancer: Current Status and Future Perspectives

**DOI:** 10.3390/jcm10194523

**Published:** 2021-09-29

**Authors:** Robert J. Walsh, David S. P. Tan

**Affiliations:** 1National University Cancer Institute, Singapore 119074, Singapore; robert_walsh@nuhs.edu.sg; 2Cancer Science Institute, National University of Singapore, Singapore 117599, Singapore

**Keywords:** cervical cancer, immunotherapy, immune checkpoint inhibitor

## Abstract

Cervical cancer remains one of the most common cancers in women around the world however therapeutic options in the advanced and recurrent setting are limited. Immune checkpoint inhibitors (ICI) have been considered an attractive option given the viral etiology of cervical cancer although the majority of patients do not benefit from their use. This review summarises current knowledge and use of immune checkpoint blockade in cervical cancer as well as discussing the challenges faced in their clinical application, namely, the role of biomarker-driven ICI use, potential mechanisms of resistance, strategies to overcome such resistance and additional immunotherapy options beyond ICI.

## 1. Introduction

Cervical cancer remains the 4th most commonly occurring cancer in women globally, despite ongoing preventive efforts with progressive human papilloma virus (HPV) vaccination programs in developed countries [1,2]. Treatment of a majority of early stage cervical cancer involves definitive chemoradiotherapy, however relapse rates at five years remain above 20% after this curative intent therapy [3].

In the setting of metastatic and recurrent cervical cancer, treatment options beyond that of first line chemotherapy combined with bevacizumab are limited, with second line single agent chemotherapy offering response rates in the region of 15–20% [4,5,6] and median survival times remaining under two years [7]. It had been hoped that the role of persistent HPV infection in cervical cancer pathogenesis [2] would leave this disease vulnerable to the effects of immune checkpoint inhibitors (ICI) however results, primarily with monotherapy ICI, have been modest to date.

Here, we discuss the current status of immunotherapy in cervical cancer including approved indications, published ICI monotherapy and combination strategies as well as immune modulation therapeutics beyond ICI. Challenges including biomarker identification and resistance mechanisms will be discussed.

## 2. Immune Response to Cancer and Immune Checkpoint Inhibition: Rationale for Immunotherapy in Cervical Cancer

The ‘cancer immunity cycle’ describes the events required for the development of an immune response to cancers [8]. Cell death leads to the release of tumour associated antigens which are phagocytosed, processed and presented via major histocompatibility complex (MHC) of antigen presenting cells (APC). Naive T-cells are activated by these non-self antigens and transit and infiltrate the tumour, with CD8^+^ T-cells and natural killer cells subsequently enacting cytotoxic cell death.

Immune checkpoints function as negative regulators of this cycle, with programmed death 1 (PD-1) and cytotoxic T lymphocyte antigen (CTLA-4) being the most studied. Tumours exploit these inhibitory pathways to evade host immune surveillance [9]. Interruption of these pathways with antibodies against targets including PD-1, its ligand programmed death ligand 1 (PD-L1) and CTLA-4 aim to facilitate a host immune response against the tumour, acting as immune checkpoint inhibitors (ICI). Use of ICI has dramatically altered the treatment landscape in many solid organ malignancies including lung, renal cell carcinoma, melanoma and colorectal cancer [10,11,12,13].

The role of HPV in the pathogenesis of cervical cancer gives insights into potential therapeutic benefit of ICI in this tumour. The majority of cases of cervical cancer are associated with high-risk HPV (16 or 18) which encode E5, E6 and E7 proteins that drive malignant transformation. These proteins are implicated in the PD1/PDL1 pathway leading to increased PD-L1 expression [14] potentially propagating immune evasion.

## 3. ICI Monotherapy and Combination in Cervical Cancer

Several studies have explored the role of ICI monotherapy and combination in advanced cervical cancer. These include the anti-PD1 (pembrolizumab, nivolumab and cemiplimab), anti-PDL1 (durvalumab) and anti-CTLA4 (Ipilimumab, tremelimumab) monoclonal antibodies. A summary of ICI monotherapy trials is shown in Table 1.

### 3.1. Pembrolizumab

KEYNOTE-028, a single arm study of Pembrolizumab in recurrent squamous cell carcinoma (SCC) cervix showed an overall response rate (ORR) of 17% in 24 PDL1 positive patients [16]. KEYNOTE-158, a basket study including a cohort of patients with advanced cervical cancer, helped to identify PDL1 as a biomarker for response. Patients (n = 98) were recruited regardless of PDL1 status and received 3 weekly pembrolizumab 200 mg for up to 2 years, with 94% being squamous histology. ORR was 12.2% in the entire study population however responses were only seen in the PDL1 positive subgroup (83.7% of study population), with PDL1 positivity determined by a combined positive score (CPS) of ≥1 based on 22C3 assay. ORR was 14.6% in the PDL1 positive group and median duration of response was not reached [17]. As a result, pembrolizumab gained US Food and Drug Administration (FDA) approval in pre-treated PDL1 positive (CPS ≥ 1) cervical cancer in 2018 [21]. 

### 3.2. Nivolumab

The anti-PD1 antibody nivolumab gave an encouraging ORR of 26.3% in preliminary analysis of Checkmate-358, a study of patients with advanced cervical cancer with ≤2 prior lines of therapy [18]. Patients with known HPV negative tumours were excluded although testing was not mandated for enrolment. PD-L1 status was assessed with 28-8 pharmDx assay with responses seen in both PD-L1 negative and positive tumours. 

A smaller study of nivolumab involving 26 patients with pre-treated advanced cervical SCC reported a lower ORR of 4%, with a further 4 patients having an unconfirmed response. The median overall survival (OS) of 14.3 months was however promising [19] and may highlight a difficulty in assessment of treatment response in the setting of immunotherapy use. 

### 3.3. Other Anti-PD-1 Agents

The EMPOWER-Cervical-1 phase III study compared cemiplimab versus physicians choice chemotherapy (pemetrexed, irinotecan, topotecan, gemcitabine, vinorelbine) in patients with advanced cervical cancer after ≥1 lines of treatment who have progressed within 6 months of platinum therapy, regardless of PD-L1 status. The primary endpoint of OS showed significant benefit for cemiplimab over chemotherapy with median OS of 12 versus 8.5 months, respectively, with benefit seen in SCC and non-SCC histology groups. Response rates were encouraging at 16.4% with estimated median DOR of 16.4 months.

AGEN2034, an anti-PD1 monoclonal antibody, showed clinical activity in phase I studies of patients with breast, ovarian and cervical cancer, with an ongoing phase II study in recurrent advanced cancer [22].

### 3.4. Anti-CTLA-4 Therapy

Monotherapy with the anti-CTLA4 agent ipilimumab showed only modest efficacy in a phase II study of both SCC and adenocarcinoma of cervix with prior exposure to platinum chemotherapy. Of 34 evaluable patients 1 showed partial response [20].

### 3.5. Anti-PD-1/PD-L1 and Anti CTLA-4 Combinations

Single agent ICI has provided encouraging but nonetheless modest results, with high rates of primary resistance observed in the studies discussed above. One possible mechanism underpinning such resistance is the concept of immune escape in both the priming and effector stages of the immune response. Priming of T-cells within lymph nodes requires not only antigen presentation but a costimulatory signal through the interaction of B7 family molecules and the T-cell expressed CD-28. CTLA-4 binds to B7 with high affinity thus inhibiting the costimulatory signal. At the tumour level immune activation leads to interferon release and subsequently an increase in PDL1 expression, thereby inhibiting T-cell response. Simultaneous blockade of both PD1/PDL1 and CTLA-4 checkpoints thus seems a rational way to attempt to increase immune response and treatment efficacy [23]. Such a combination (anti-PD1/PDL1 and anti-CTLA4) has been shown to be effective in several tumour groups leading to approvals in the first line advanced setting in mesothelioma [24], renal cell carcinoma [25], melanoma and non-small cell lung cancer (NSCLC) [25]. This approach has been explored in advanced cervical cancer.

Checkmate-358 reported outcomes for nivolumab and ipilimumab administered at 2 different dose combinations: Nivolumab 3 mg/kg 2 weekly with ipilimumab 1 mg/kg 6 weekly (combination A) and nivolumab 1 mg/kg with ipilimumab 3 mg/kg 3 weekly for 4 doses followed by nivolumab maintenance 2 weekly (combination B). The response rate was higher with combination B versus A with ORR of 41.3 and 26.7%, respectively. Responses were also seen in patients with PD-L1 negative tumours (2/14, 14.3% combination A; 4/11 36.4%, combination B). Survival at 12 months in patients without prior treatment was encouraging in both arms with rates of 83.5 and 78% for combination A and B, respectively [26].

Results of two trials of balstilimab (anti-PD1) alone and in combination with zalifrelimab (anti-CTLA-4) in patients with advanced cervical cancer (SCC or adenocarcinoma) progressing after platinum therapy, were presented at ESMO 2020 with higher response rates seen in the trial of combination therapy [27]. ORR in the single arm study of balstilimab alone was 14% with 10% of PDL1 negative tumours showing treatment response. ORR of 22% was seen in the combination trial of balstilimab and zalifrelimab with responses again seen in both PDL1 positive (ORR 27%) and PDL1 negative tumours (ORR 11%). Of note, responses were seen in both SCC and adenocarcinoma with both single agent balstilimab and combination therapy.

The anti-PDL1 antibody durvalumab in combination with tremelimumab had limited activity in a phase I study with no responses seen in cervical cancer patients although stable disease more than 24 weeks was seen in 15.4% [28]. Other combinations under investigations include AGEN1884 (anti-CTLA-4) and AGEN2034 (anti-PD-1), with safety data reported from a phase I/I trial including patients with refractory solid organ malignancies (expansion in patients with cervical cancer) [29].

### 3.6. Role for ICI in Small Cell Neuroendocrine Carcinoma of the Cervix

Neuroendocrine carcinoma of the cervix (NECC) is an uncommon histology reported in 1.4% cervical cancers [30]. The most common variant is small cell NECC which runs an aggressive course and has poor prognosis. The optimal treatment strategy is unclear as is the efficacy of ICI with large studies currently lacking, however case reports point to potential role, with nivolumab resulting in radiologic complete response in one patient with metastatic PDL1 negative small cell NECC [31].

Analysis showing a high rate of HPV positivity suggests a viral role in the aetiology of NECC as seen in cervical SCC [32]. This viral pathogenic factor provides reasoning for further investigation of the use of ICI therapy in this histological subgroup. Furthermore, in first line treatment of the morphologically similar small cell lung cancer (SCLC) anti-PDL1 therapy has been shown to improve OS when used in combination with platinum-based chemotherapy in two phase III studies [33,34]. However, in pre-treated SCLC single agent ICI shows only modest results contributing to the voluntary withdrawal of approval for nivolumab for use in treatment refractory SCLC [35].

## 4. Challenges of ICI Therapy in Cervical Carcinoma

Results of studies discussed above highlight that while patients with cervical cancer can gain benefit from the use of ICI therapy, responses are seen in the minority. Early reports of combination ICI appear to show improved response rates however further confirmatory data from randomised studies are awaited. Key to improving ICI efficacy is to identify potential biomarkers for response as well as considering ways to overcome immunotherapy resistance. Currently available biomarkers and selected strategies to overcome such resistance, including combining ICI with chemotherapy, radiotherapy and anti-angiogenics, are discussed below. Selected ongoing combination studies in early/locally advanced and recurrent/metastatic cervical cancer are highlighted in Table 2 and Table 3, respectively.

### 4.1. Role of Biomarkers

PDL1 is not expressed in normal cervical tissue [36] but is seen in malignant and pre-malignant lesions with reported rates of 95% and 80% in cervical intraepithelial neoplasia (CIN) and cervical SCC, respectively [36], while another study reported a low PDL1 expression rate of 24.9% in cervical SCC [37]. Reported expression levels vary partly due to the use of differing assays and cut-offs for positivity used. Rates of expression are lower in cervical adenocarcinoma compared to SCC with one study reporting PDL1 tumour cell positivity in 14% adenocarcinoma samples versus 54% SCC [38].

PDL1 is an FDA approved biomarker in cervical cancer with positive tumours eligible for single agent pembrolizumab after progression on first line chemotherapy based on results of KEYNOTE-158 discussed above [17]. PDL1 positivity was determined by a CPS ≥ 1 using IHC 22C3 pharmDx assay, with CPS being the sum of PDL1-stained cells (tumour cells, lymphocytes, and macrophages) divided by the total number of viable tumour cells, multiplied by 100 [39]. Use of CPS positivity to identify those cancers who may benefit from PD1/PDL1 blockade would therefore seem appropriate, however other studies have reported responses in PDL1 negative tumours, including Checkmate-358 which used the IHC 28-8 pharmDx assay [18]. Whether these differences are a result of the different assays used, or that truly PDL1 is not a robust biomarker in cervical cancer is not clear. Heterogenicity in PDL1 expression may contribute to differing results regarding its role as a biomarker, with an analysis of paired primary and metastatic samples showing discordant tumour cell PDL1 expression in 31% (22/71) of cases [38]. Alternative methods of PD1/PDL1 assessment beyond IHC have been assessed but are not in widespread use. Reports of PDL1 copy number analysis show the rate of amplification to be low, being seen in 2% of cervical SCC cases and 0.7% of a cohort of various solid tumour types making it a poor candidate [40,41]. Use of RNAish to detect PDL1 mRNA appears promising, with expression in 56% of tumour cells reported [41].

Tumour mutational burden (TMB) is calculated by assessing the number of non-synonymous somatic mutations per mega-base (mb) and is a surrogate for tumour neoantigen load and potential immunogenicity [42,43]. A study of 284 cervical SCC specimens showed a median TMB of 5.4 mutations/mb, with 6.7% cases exhibiting TMB > 20 mutations/mb [44]. Studies in lung cancer have suggested improved efficacy of ICI with higher TMB [45] and biomarker analysis of the basket study KEYNOTE 158 led to the approval of pembrolizumab in patients with high TMB (≥10 mutations/mb) based on the companion diagnostic FoundationOne CDx assay. This analysis included 16 patients with cervical SCC with an ORR 31% [46]. While this is promising and offers another avenue for patients to access ICI response rates remain modest.

Microsatellite instability or deficiency in mismatch repair proteins also represents a tumour agnostic indication for the use of pembrolizumab in refractory solid organ malignancies. In a study of 93 cases of cervical SCC, microsatellite instability-high was reported in 11.8%, with a lower rate of 3.6% reported in a separate study (n = 168) [47,48].

A study of tumour draining lymph node and primary tumour samples identified CD8^+^FoxP3^+^CD25^+^ effector T cells as a potential alternative biomarker for efficacy of PD-1/PDL1 blockade that merits investigation, with an association seen between the percentage this T cell subset and IFNγ response after PD-1 inhibition in vitro using single cell suspensions [49]. 

As molecular biomarkers continue to be developed it is important to assess the role of clinical biomarkers in identifying those who may gain the most from ICI use. Retrospective studies including various tumour types have highlighted overweight and obese patients to have improved response rates versus non-overweight patients [50], possibly driven by immune dysregulation associated with obesity in pre-clinical models [51]. In NSCLC primary resistance to ICI therapy was seen to be associated with factors including never smokers, more involved sites, more prior treatments, and lower mean albumin [52]. Currently such clinical biomarkers to not play a role in treatment decisions and limited data is available for cervical cancer patients where further research is needed. 

### 4.2. Resistance Mechanisms and Treatment Strategies

Resistance mechanisms to ICI therapy in cervical cancer are not well described. Work in lung cancer and melanoma has begun to highlight key tumour and host factors in such resistance to immunotherapy [53].

#### 4.2.1. Immunosuppressive Microenvironment

Dysregulation of microenvironment can create an immunosuppressive, ‘cold’ and non-inflamed tumour. A study of 40 patients highlighted a change from Th1 to immunosuppressive Th2 state as pre-malignant lesions progressed from CIN-1 to CIN-3 [54]. Conversely, ‘hot’ inflamed tumours are associated with T cell infiltration, with tumour infiltrating lymphocytes (TILs) linked to improved survival [55,56]. Analysis of surgical specimens from 86 patients with FIGO I-II cervical SCC showed intra-epithelial M1 macrophages was associated with improved survival as well as high numbers of TILs [57]. 

An immunosuppressive microenvironment was seen in tumour draining lymph nodes involved by cervical carcinoma compared to lymph nodes free of tumour with higher number of CD4 and CD8 positive Tregs and increased expression of PDL1 and B7-H4, a coinhibitory molecule [58]. In vitro analysis highlighted an immunosuppressive cytokine profile in tumour involved lymph nodes consisting of higher levels of IL6, IL10, and TNFα released under stimulation, while IFNγ release was high in cells from tumour free lymph nodes. The phase 1 DURVIT study assesses the safety of locally administered durvalumab in patients with cervical cancer planned for hysterectomy and lymph node dissection with key secondary endpoints assessing effect on the microenvironment of both the tumour and draining lymph nodes [59]. 

Mutations in PIK3CA have been associated with an immunosuppressive microenvironment and were reported in 40% patient patients in a prospective analysis of treatment naive cervical cancer [60]. When present in tandem with loss of function mutations in epigenetic pathway regulators (34% of cases) PIK3CA mutations were associated with significantly shorter PFS. Mouse tumour models with a common PIK3CA mutation (H1047R) showed reduced infiltrate of CD8+ T cells and resistance to immunotherapy (anti PD-1) that could be reversed with PI3K inhibition [61]. Studies in melanoma also highlight role of PIK3-AKT pathway activation in immune resistance. Tumours with PTEN loss showed inferior reduction in tumour size versus those with retained expression after treatment with nivolumab or pembrolizumab, while a combination PI3Kβ inhibition and anti-PD1 therapy in mouse models of PTEN null melanoma achieved improved tumour control versus either agent alone [62]. An ongoing phase I trial is examining the role of combining the AKT inhibitor AZD5363 with durvalumab and olaparib in patients with treatment refractory solid organ malignancies with PIK3-AKT pathway mutations (NCT03772561).

#### 4.2.2. Role of VEGF Signalling

Vascular endothelial growth factor (VEGF) is associated with immunosuppressive microenvironment and reduced lymphocyte influx in a number of tumours [63]. It is seen to prevent maturation of dendritic cells while leading to increased numbers of inhibitory Tregs and tumour associated macrophages [64]. Additionally, in vitro, VEGF-A appears to enhance expression of PD-1 on CD-8 T cells thereby potentiating immunosuppressive signals [65]. Inhibition of VEGF signalling is associated with ‘normalisation’ of tumour vessels and CD8+ T cell response [63] and the combination of ICI and VEGF inhibition has become an attractive option to examine in clinical trials. Such combination strategies have shown a survival benefit in NSCLC [66], hepatocellular carcinoma [67] and renal cell carcinoma [68]. Anti-VEGF therapy is appealing in cervical cancer with upregulation of VEGF-A and VEGF receptor 1 seen in cases of recurrent disease [69] and addition of bevacizumab (anti-VEGF-A) to chemotherapy is currently first line standard of care for recurrent/metastatic cervical cancer in view of reported survival benefit over chemotherapy alone [7].

The CLAP study, a single arm phase II trial of anti-PD1 camrelizumab in combination with apatinib showed an impressive ORR of 55.6% in patients with pre-treated advanced cervical cancer [70]. In total. 33% of patients were PDL1 negative or unknown with response seen in both PDL1 positive and negative cases. Ongoing phase III studies will inform regarding the role of such combinations in the first line setting. The first line study KEYNOTE 826 is assessing the addition of pembrolizumab or placebo to the combination of chemotherapy with or without bevacizumab (investigators choice), while the first line BEATcc study involves chemotherapy, bevacizumab with or without atezolizumab [71].

#### 4.2.3. Tumour Antigen Presentation

Disruption of the antigen presentation pathway is seen across multiple tumour types as a mechanism of immune evasion and ICI resistance [72]. Mutations in β_2_-microglobulin mutations affect MHC functioning and are associated with ICI resistance in melanoma and NSCLC [73,74]. Thiol reductase ERp57 has a role in MHC assembly, and its downregulation in cervical cancer and associated with worse OS [75,76].

Combinations including chemotherapy and/or radiotherapy can work to increase immunogenic cell death leading to the release of tumour associated neoantigens and cellular danger-associated molecular patterns, resulting in increased activity of APC and downstream T cell activation. Combining ICI with chemoradiotherapy is an attractive option with ongoing studies in the locally advanced setting including KEYNOTE-A18, a phase III trial evaluating the addition of pembrolizumab to standard of care concurrent chemoradiotherapy (NCT04221945). Table 2 highlights selected studies trialling similar combinations of ICI and concurrent chemoradiotherapy. 

#### 4.2.4. Co-Inhibitory Signalling Pathways

T cell immunoglobulin and mucin-domain-containing molecule 3 (TIM-3) negatively regulates immune response. TIM-3 is co-expressed with PD-1 on CD-8+ T cells and is associated with T cell exhaustion with TIM3+ CD4 T cells producing less interferon-γ and IL-2 than TIM3 negative cells [77,78]. In a study of 42 cervical SCC specimens TIM3 expression correlated with tumour grade and presence of metastasis [79]. Early phase trials are ongoing evaluating the role of TIM-3 inhibition alone and in combination with anti-PD1/PDL1 therapy in patients with advanced solid organ cancers (NCT03652077, NCT02608268).

Upregulation of additional immune checkpoints including lymphocyte-activation gene 3 (LAG-3) has been seen in multiple cancers including cervical cancer and is thought to play a role in adaptive resistance to ICI [80,81]. DUET-4 trial of the bispecific antibody targeting CTLA-4 and LAG-3 alone or in combination with pembrolizumab is an early phase study of patients with advanced malignancies including cervical cancer (NCT03849469).

T cell immunoglobulin and ITIM domain (TIGIT) binds to ligands CD155 and CD 112 exerting inhibitor signals on T cell response. Co-expression has been seen with other immune checkpoints in cervical cancer [82,83]. The ongoing SKYSCRAPER 04 examines the combination of atezolizumab alone and in combination with the anti-TGIT tiragolumab in patients with advanced cervical cancer after 1–2 prior lines of chemotherapy (NCT04300647).

## 5. Immunotherapy beyond ICI 

Aside from ICI there are numerous immunotherapy strategies under investigation, including several in the setting of advanced cervical cancer discussed below. 

### 5.1. Cancer Vaccines

The association of HPV and cervical SCC makes HPV related proteins attractive targets for vaccine based therapy. The vector vaccine ADXS11-001 is an attenuated live Listeria monocytogenes encoding the E7 oncoprotein. Initial results of the GOG/NRG0265 study of ADXS11-001 are promising with a 12 month OS rate of 38.5% in patients with pre-treated recurrent or metastatic cervical carcinoma (squamous and non-squamous) [84]. A combined 12 month OS rate of 34.9% was seen in a phase II study of ADXS11-001 with and without cisplatin in patients with advanced cervical carcinoma [85].

Peptide vaccines are also under investigation. ISA101 consists of 12 synthetic long peptides from the E6/7 oncoproteins of HPV 16 and is combined with nivolumab in patients with advanced HPV16 positive tumours in a single arm phase II study (NCT02426892) including 1 patient with cervical cancer. ORR in the overall population was 33% [86]. 

### 5.2. Genome Editing Tools

Research utilising clustered regularly interspaced palindromic repeats (CRISPR) associated protein 9 (Cas9) technology to enact genetic editing is a rapidly growing field [87].

Transcription Activator-Like Effector Nucleases (TALENs) also function as gene editing tools and TALEN targeting E7 oncoprotein is seen in vitro to downregulate E7 expression and lead to cell death [88]. A phase I study of CIN patients will evaluate TALEN-HPV E6/E7 and CRISPR/Cas9-HPV E6/E7 (NCT 03057912).

### 5.3. Cell Based Therapy—Engineered T Cells

Adoptive cell transfer is promising field with a phase II study involved patients receiving lymphodepletion with cyclophosphamide and fludarabine followed by infusion of TILs (LN-145), and up to 6 doses of IL-2 showed an impressive ORR of 44% in patients with advanced cervical cancer progressing on prior chemotherapy [89].

## 6. Conclusions

Significant challenges exist in our efforts to improve responses to ICI in cervical cancer. One of the key deficiencies remains the lack of more robust biomarkers beyond PD-L1 expression, high TMB or MSI-high. Even in tumours with these established biomarkers, response rates remain modest, but importantly long-lived in the minority of responders. 

Encouraging improvements in efficacy have been observed with early reports of ICI doublet (anti-PD1/PDL1 and anti-CTLA4) and ICI plus anti-angiogenic therapy. Rationally designed clinical trials incorporating biomarker discovery with combination strategies, including the addition of PARP inhibitors, therapeutic vaccines and radiotherapy to ICI, will be crucial in unravelling the various mechanisms of ICI resistance that exist within cervical cancer. More importantly, the answers to these questions will be the key to expanding the role of ICIs in cervical cancer.

## Figures and Tables

**Table 1 jcm-10-04523-t001:** Reported trials of ICI single agent therapy in advanced cervical cancer.

Trial	No. of Subjects	Included Subjects	Intervention	ORR (%)	mDOR (Months)	Survival (Months)
EMPOWER-CERVICAL 1/GOG-3016/ENGOT-CX9 [15]	608	PDL1 unselected ≥1 prior line of therapy	Cemiplimab 350 mg 3 weekly vs. Investigator choice chemotherapy	16.4 vs. 6.3	16.4 vs. 6.9	mPFS: 2.8 vs. 2.9 (HR = 0.75 [0.63–0.89]) mOS: 12.0 vs. 8.5 (HR = 0.69 [0.56–0.84])
KEYNOTE-028 [16]	24	PD-L1 ≥ 1% (modified proportion score) PD on prior therapy	Pembrolizumab 10 mg/kg, 2 weekly	17	5.4 m	mPFS 2 mOS 11
KEYNOTE-158 [17]	98	PDL1 unselected PD on prior therapy	Pembrolizumab 200 mg 3 weekly	12.2 (PDL1 unselected) 14.6 (PDL1 CPS ≥ 1)	NR	mPFS 2.1 mOS 9.4 (ITT)
CHECKMATE-358 [18] cervical cohort	19	≤2 prior lines of therapy Excluded HPV negative	Nivolumab 240 mg 2 weekly	26.3	NR	mPFS 5.1 mOS 21.9
NRG-GY002 [19]	25	PDL1 unselected PD on prior therapy	Nivolumab 3 mg/kg 2 weekly	4	3.8	mPFS 3.5 mOS 14.5
Lheureux et al. [20]	42	PDL1 unselected	Ipilimumab 10 mg/kg 3 weekly (4 cycles) → 12 weekly maintenance (to 1 year)	3	-	mPFS 2.5 mOS8.5

mDOR, median duration of response; ORR, objective response rate; HR, hazard ratio; mOS, median overall survival; mPFS, median progression free survival; PD, progressive disease; m, months; NR, not reached.

**Table 2 jcm-10-04523-t002:** Selected ongoing trials—Early/locally advanced cervical cancer.

Trial Identifier	Study Phase	Study Population	Intervention
NCT04221945	III	FIGO 2014 IB2-IIB (node positive), FIGO 2014 II-IVA	CRT vs. CRT + concurrent and adjuvant pembrolizumab
NCT03830866	III	FIGO (2009) Stages IB2 to IIB node positive or FIGO (2009) IIIA-IVA any nodal status	CRT vs. CRT + durvalumab then durvalumab maintenance (2 years)
NCT02635360	II	-	CRT + concurrent Pembrolizumab vs. CRT adjuvant pembrolizumab
NCT03527264	II	FIGO 1B-IVA	1A: CRT + concurrent Nivolumab (whole pelvic RT) 1B: CRT + concurrent Nivolumab (extended field RT) 2: CRT + Nivolumab maintenance (total 2 years) 3: CRT + concurrent Nivolumab then maintenance Nivolumab (total 2 years)
NCT03612791	II	FIGO IB2-IVB (limited to PA nodes)	CRT vs. CRT + concurrent atezolizumab (atezolizumab total 20 cycles)
NCT03833479	II	FIGO IB2/IIA2/IIB (positive pelvic LN) FIGO IIIA/IIIB/IVA	CRT + consolidation TSR-042 (2 years)
NCT04238988	II	FIGO IB2-IIB	Neoadjuvant carboplatin + paclitaxel + pembrolizumab → surgery → Adjuvant carboplatin-paclitaxel-pembrolizumab (high-risk patients)
NCT01711515	I	FIGO (2014) IB2/IIA (+para-aortic LN), IIB/IIIB/IVA	CRT + adjuvant Ipilimumab
NCT04256213	Pilot	FIGO IB3-IVA	Ipilimumab + Nivolumab + CRT

CRT, chemoradiotherapy; LN, lymph node; PA, para-aortic; RT, radiotherapy.

**Table 3 jcm-10-04523-t003:** Selected ongoing trials in advanced and metastatic cervical cancer.

Trial Identifier	Study Phase	Treatment Status	Study Population	Intervention
**ICI + Chemotherapy**				
NCT03635567	III	Naive	Recurrence/metastatic cervical cancer	Cisplatin/Carboplatin + Paclitaxel + Bevacizumab + Pembrolizumab/placebo
NCT03556839	III	Naive	Stage IVB, persistent/recurrent cervical cancer	Cisplatin/Carboplatin + Paclitaxel + Bevacizumab +/− Atezolizumab
NCT03340376	II	Pre-treated	Recurrent/metastatic cervical cancer	Atezolizumab vs. Doxorubicin vs. Atezolizumab + Doxorubicin
NCT03518606	I/II	Pre-treated	Recurrent/metastatic Cervical, H+N, Breast, Prostate cancer	Durvalumab + Tremelimumab + Vinorelbine
NCT04188860	II	Pre-treated	Recurrent/persistent advanced cervical cancer	Camrelizumab + Nab-paclitaxel
**ICI + Targeted Therapy**				
NCT03826589		Pre-treated	Recurrent/metastatic cervical cancer	Avelumab + Axitinib
NCT04357873	II	Naive/pre-treated	Recurrent/Metastatic SCC (Vulvar, Penile, Cervix, H + N, Anal)	Pembrolizumab + Vorinostat
NCT04230954	II	Naïve (PD-L1 CPS ≥ 1)	Recurrent/Metastatic Cervical cancer	Pembrolizumab + Cabozantinib
NCT04483544	II	≤2 prior lines		Pembrolizumab + Olaparib
**ICI + Radiotherapy**				
NCT03614949	II	Naive/pre-treated	Recurrent/metastatic Cervical cancer or HPV positive SCC of vagina/vulva.	Atezolizumab + SBRT (24Gy, 3 fractions)
NCT03277482	I	Pre-treated	Recurrent/metastatic gynaecological cancer	Durvalumab, Tremelimumab + Radiation therapy
**Vaccine Therapy +/− ICI**				
NCT03946358	II	Pre-treated	Pre-treated locally advanced/metastatic HPV associated cancers	Atezolizumab + UCPVax
NCT04405349	IIa	Pre-treated	HPV16 + ’ve cervical Ca	VB10.16 + Atezolizumab
NCT03073525	II	-	Advanced gynaecological malignancy	Part 2:Vigil x2 → Vigil + Atezolizumab Part 2 comparator: Atezolizumab x2 → Atezolizumab + vigil
NCT04432597	I/II	Naive/pre-treated	Recurrent/metastatic HPV associated cancer	PRGN-2009 +/− M7824
NCT02866006	I/II	Pre-treated	Recurrent/metastatic HPV 16/18 positive Cervical cancer	BVAC-C
NCT02128126	I/II	Naive	Recurrent/metastatic cervical cancer	ISA101/ISA101b + Carboplatin + paclitaxel +/− Bevacizumab
NCT04287868	I/II	Pre-treated	Advanced HPV associated malignancies	PDS0101 + M7824 + NHS-L12

HPV, human papilloma virus; H + N, head and neck; SBRT, stereotactic body radiation therapy.

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
