# Peer review of "The Role of Immunotherapy in the Treatment of Advanced Cervical Cancer: Current Status and Future Perspectives"

_jcm, 2021, doi:10.3390/jcm10194523_

Round 1

Reviewer 1 Report

Please find the review in attachement.

Author Response

Dear Reviewer, 

Many thanks for your review and suggestions on the subject of clinical biomarkers. We have amended the manuscript to include the below paragraph in section 4.1 Role of biomarkers:

As molecular biomarkers continue to be developed it is important to assess the role of clinical biomarkers in identifying those who may gain the most from ICI use.  Retrospective studies including various tumour types have highlighted overweight and obese patients to have improved response rates versus non-overweight patients [46], possibly driven by immune dysregulation associated with obesity in pre-clinical models[47]. In NSCLC primary resistance to ICI therapy was seen to be associated with factors including never smokers, more involved sites, more prior treatments, and lower mean albumin[48].  Currently such clinical biomarkers to not play a role in treatment decisions and limited data is available for cervical cancer patients where further research is needed.

Once again, thank you for the time taken to review and offer comments to help strengthen our article. 

Reviewer 2 Report

After having read this review paper I am convinced that it adds valuable and comprehensive information in depth about the current status and challenges of the role of immunotherapy and immune biomarkers in advanced cervical cancer.

Despite enormous efforts in the past, effective therapy of the advanced cervical cancer is still unresolved. Although the expectations are on high level from the clinicians for immunotherapy, the authors  moderately and censoriously present the current knowledge of the topic and the limitations. 

The paper is well written and structured, supported by tables and robust amount of references.

It offers a good summary for those are not experienced in clinical gynecologic oncology.

In order to make it more enjoyable I have one comment: a list of abbreviations would help the smooth understanding. 

Author Response

Dear Reviewer, 

Many thanks for taking the time to review our article and offer comments in order to strengthen it.  We have amended the article to include a list of abbreviations used below the Keywords section and will be guided by the editor on the final location and format of this list.  

See attached for the current format of the suggested list. 

Yours faithfully, 

Dr Robert Walsh

Reviewer 3 Report

This is an instructivem, professional and scientifically provocative work.

Author Response

Dear Reviewer, 

Many thanks for the time taken to review our article and offer comments.  We appreciate your feedback. 

Yours faithfully, 

Dr Robert Walsh